# Electrochemical Investigation of Phenethylammonium Bismuth Iodide as Anode in Aqueous Zn^2+^ Electrolytes

**DOI:** 10.3390/nano11030656

**Published:** 2021-03-08

**Authors:** Stylianos Daskalakis, Mingyue Wang, Claire J. Carmalt, Dimitra Vernardou

**Affiliations:** 1Department of Electrical and Computer Engineering, School of Engineering, Hellenic Mediterranean University, 71410 Heraklion, Greece; s.f.daskalakis@gmail.com; 2Christopher Ingold Laboratory, Department of Chemistry, University College London, 20 Gordon Street, London WC1H 0AJ, UK; mingyue.wang@ucl.ac.uk (M.W.); c.j.carmalt@ucl.ac.uk (C.J.C.); 3Institute of Emerging Technologies, Hellenic Mediterranean University Center, 71410 Heraklion, Greece

**Keywords:** AACVD, organohalides, anodes, Zn-ion, intercalation performance

## Abstract

Despite the high potential impact of aqueous battery systems, fundamental characteristics such as cost, safety, and stability make them less feasible for large-scale energy storage systems. One of the main barriers encountered in the commercialization of aqueous batteries is the development of large-scale electrodes with high reversibility, high rate capability, and extended cycle stability at low operational and maintenance costs. To overcome some of these issues, the current research work is focused on a new class of material based on phenethylammonium bismuth iodide on fluorine doped SnO_2_-precoated glass substrate via aerosol-assisted chemical vapor deposition, a technology that is industrially competitive. The anode materials were electrochemically investigated in Zn^2+^ aqueous electrolytes as a proof of concept, which presented a specific capacity of 220 mAh g^−1^ at 0.4 A g^−1^ with excellent stability after 50 scans and capacity retention of almost 100%.

## 1. Introduction

The ever-growing demand for plug-in electric vehicles and smart grids require the development of batteries with high energy and power density at low manufacturing costs and long life cycles. The development of a technology that circumvents the limitations of up-to-date batteries is aqueous batteries, which hold advantages when compared to non-aqueous batteries since they do not require oxygen- or water-controlled manufacturing environments. Li-ion batteries have attracted considerable attention in both commercial markets and academic studies of various battery systems [1,2]. However, lithium still suffers from a low energy density of <350 Wh kg^−1^ based on the intercalation chemistry, potential safety risk, and limited natural resources to meet the energy-storage demands for large-scale applications [3,4]. Therefore, there has been increasing attention on multivalent ions, such as Zn^2+^, in recent years because they can achieve higher storage capacity and energy density due to the multiple electron transfer involved in redox reactions [5,6,7]. In particular, Zn-ion batteries are promising due to their large theoretical energy density (1353 Wh kg^−1^, excluding oxygen), low cost, high electrical conductivity, inherent safety [8,9], and electrochemical stability in water due to high overpotential for hydrogen evolution [10]. Nevertheless, Zn-ion batteries suffer from zinc precipitation, zinc anode dendrite formation, limited O_2_ solubility in electrolytes, and higher charge overpotential [11]. Specifically, dendrite growth upon recharge may penetrate the separator and reach the cathode, leading to a catastrophic failure of the battery [4]. Several strategies have been studied to tackle these challenges, including (a) high surface area Zn anodes (i.e., fibers, sponges, and foams) to enhance the electrochemical performance, (b) surface modifications with additives to suppress H_2_ evolution, (c) the incorporation of alloying metals to increase conductivity and improve the current distribution, (d) surface trapping layers for better retention of the soluble zincate discharge product, and (e) wrapping Zn with thin protective layers [4]. Research activity has also been directed at the design of anode electrodes with high specific capacity, safety, and stability performance that are higher than those of the present commercial electrodes on the market [12].

Potential anodes include graphite, however, the low charging rate and theoretical specific capacity of 372 mAh g^−1^ restricts the further development of their commercialization [13]. Among the alternative active materials, Si has attracted attention due to the high theoretical specific capacity being 4200 mAh g^−1^. However, there are issues related to volume expansion during the intercalation/deintercalation processes, limiting its application in batteries [14,15]. Additionally, transition metal oxides such as Fe_2_O_3_ [16], MnO_2_ [17], and Co_3_O_4_ [18,19] have been reported as anode materials due to their high abundance, low cost, and divergent flexible oxidation states. However, there are issues with toxicity in some of these materials and low electrical conductivity, which gives rise to structural collapse of the material and leads to poor stability [20].

A promising class of materials includes halide perovskites, which have received great attention since 2009 [21] due to their excellent optoelectronic performance in solar cells with superior power conversion efficiencies of 25.2% [22], field effect transistors (FETs) [23], light-emitting devices (LEDs) [24], and photodetectors [25]. These materials combine the properties of inorganic (high carrier mobility and a wide range of bandgaps) and organic materials (structural diversity, high efficiency luminescence, and plastic mechanical properties [26]) [27]. There are studies that utilize organic–inorganic materials as charge storage materials for Li^+^ battery anodes, which, however, show poor performance (i.e., specific capacity of 50 mAh g^−1^) [28]. In addition, the toxicity of lead and the instability in the presence of moisture and oxygen are prime concerns preventing their commercialization [29,30].

Methylammonium bismuth iodide (MA_3_Bi_2_I_9_) consists of a dimeric anionic structure with two face-sharing octahedra (Bi_2_I_9_^3−^). This material has been studied before as an electrochemical double-layer capacitor with excellent stability (i.e., retained 84.8% of its initial maximum capacitance even after 10,000 charge/discharge cycles) [31]. Furthermore, it was shown that the stability of the active perovskite layer could be further improved through the replacement of MA^+^ with a larger organic ligand, such as the phenethylammonium cation (PEA^+^), indicating excellent thermal stability and resistance against moisture [32,33]. However, until now the application of a MA_3_Bi_2_I_9_ or (PEA)_3_Bi_2_I_9_ anode has not been studied in aqueous Zn-ion batteries.

In this paper, we report the electrochemical properties of phenethylammonium (C_8_H_12_N^+^, PEA) bismuth iodide [33] in an aqueous solution of ZnSO_4_^·^7H_2_O. For this purpose, the stability window was identified, which is very crucial in aqueous electrolytes [34]. The (PEA)_3_Bi_2_I_9_ was aerosol-assisted chemical vapor deposited (AACVD) at atmospheric pressure. Through AACVD [35], the large-scale production of electrodes at a competitive cost can be achieved, especially under an atmospheric pressure operation. In addition, a low-cost approach using an aqueous solution of ZnSO_4_ rather than organic electrolytes can withstand overcharge, simplifying the charging equipment [36].

## 2. Materials and Methods

### 2.1. AACVD Procedure for the Growth of (PEA)_3_Bi_2_I_9_

A 4 mL 0.1 M (PEA)_3_Bi_2_I_9_ precursor solution in dimethylformamide (DMF) prepared in situ was used to deposit the film in a cold-wall Chemical Vapour Deposition reactor. The precursor solution was placed in a glass bubbler and an aerosol mist was generated using a piezoelectric device. All chemicals were purchased from Merck (Boston, MA, USA). Depositions were carried out under nitrogen at a flow rate of 0.8 L min^−1^. The reactor was built up in a top-down heating configuration as previously reported [37]. The carbon block was positioned above a metal plate, which supported the fluorine doped tin dioxide (FTO) glass substrate (13 cm × 4.5 cm × 0.3 cm) parallel to the carbon block. The temperature of the carbon heating block was 400 °C during the film deposition process. The whole set was enclosed within a quartz tube.

### 2.2. Material Characterization

Gradient-incident X-ray diffraction (GIXRD) patterns were obtained using a Bruker-Axs D8 diffractometer (Cambridge, UK) with parallel beam optics and a position sensitive detector (PSD) LynxEye silicon strip detector. This instrument used monochromatic Cu K_α1_ radiation (λ = 1.5406 Å) at 40 kV with a 30 mA emission current, and the incident beam angle was set to 1°. The film morphology was investigated by a Japan Electron Optics Laboratory (JEOL) JSM-6301F field emission scanning electron microscope (SEM) (Herts, UK). The sample was sputtered with a thin layer of gold to improve the surface electrical conductivity in order to avoid charging effects.

Electrochemical studies of the sample were accomplished utilizing a three-electrode electrochemical cell (Easycon Hellas, Ioannina, Greece) as reported in [38]. In particular, Pt, Ag/AgCl, and (PEA)_3_[Bi_2_I_9_] were the counter, reference, and working electrodes, respectively. An aqueous solution of 1 M ZnSO_4_·7H_2_O (i.e., ZnSO_4_ + H_2_O -> Zn^2+^ + SO_4_^2−^) acted as the electrolyte for a potential range of −0.5 V to +1.5 V. The mass of the anode electrode was estimated by a 6-digit analytical grade scale measuring the FTO glass substrate before and after the growth. It was found to be 0.000012 g.

Using analogous Zn^2+^ aqueous solutions such as ZnSO_4_, ZnCl_2_, Zn(CH_3_COO)_2_ and Zn(NO_3_)_2_, it was found that SO_4_^2−^ ions had a higher stability and compatibility than other anions achieving better reversibility and faster Zn-ion kinetic behavior [39]. This outcome led to that particular electrolyte being utilized in our work.

## 3. Results and Discussion

### 3.1. Materials Characteristics

(PEA)_3_Bi_2_I_9_ films were deposited onto FTO glass substrates via AACVD of (PEA)_3_Bi_2_I_9_ in DMF at 400 °C. The films showed excellent coverage across the substrate and were uniform. A comparison of XRD patterns for the film deposited and the calculated pattern are shown in Figure 1a. The peaks in the GIXRD pattern of the film were in good accordance with the values in the literature regarding monoclinic space group *P2_1_/n* (PEA)_3_Bi_2_I_9_ [33]. In addition, a preferential orientation toward (1, 0, −1) existed in the film growth. As observed in the SEM image (Figure 1b), a film with a compact arrangement of crystal grains was deposited via AACVD, and the grain size was around 1 μm.. The surface morphology of the (PEA)_3_Bi_2_I_9_ film was rough due to the island growth in the AACVD process.

### 3.2. Electrochemical Studies

A series of experiments was conducted concerning the electrolyte concentration. The anode electrode was initially tested in a range of 0.5 M to 0.8 M, with ZnSO_4_^·^7H_2_O showing cyclic voltammetry curves with noise (i.e., very low current density to the order of 10^−8^ A). Continuing our efforts with higher molar concentrations up to 2 M, the current density was higher and similar for the particular range of values. However, the solution was blurred for the highest concentration value (i.e., 2 M) and it was not possible to observe any changes taking place in the electrode during the intercalation/deintercalation scans. Hence, the optimum concentration value for the particular experiments was found to be 1 M ZnSO_4_·7H_2_O.

Regarding the estimation of the electrochemical window, we initially started our experiments with a narrow window and slowly increased it, reaching −0.5 V to +1.5 V and showing good stability. However, by increasing it further to +1.8 V, the anode electrode material fell apart and it was not possible to continue the measurement.

Figure 2a shows the cyclic voltammetry (CV) curves of (PEA)_3_Bi_2_I_9_ at a scan rate of 10 mV s^−1^ for 1, 100, and 300 scans. The first scan shows one cathodic peak at +1.35 V (Zn^2+^ deintercalation) and two anodic peaks at +0.88 V and −0.27 V (Zn^2+^ intercalation). The current density gradually increased from the first to the 100th scan due to activation of the anode [40]. Following the 100th scan, the curves almost overlapped, indicating good electrochemical stability of the anode. The peak positions remained almost unchanged for higher scan numbers, presenting a reversible phase transition during the Zn^2+^ intercalation/deintercalation processes.

In order to evaluate the Zn^2+^ kinetics in (PEA)_3_Bi_2_I_9_, the CV curves were measured at various scan rates ranging from 1 to 30 mV s^−1^ with a potential window of −0.5 V to +1.5 V (Figure 2b). With a continuous increase in the scan rates, the shape of the curves remained unchanged, but the cathodic and anodic peaks gradually shifted to higher and lower voltages owing to the polarization effect [41]. The plot of the peak current against the square root of the scan rate shows that a linear relationship was obtained for the cathode (at +1.33 V) and the anode (at +0.86 V), demonstrating that reaction kinetics were controlled by diffusion processes (Figure 2c). In that case, the diffusion coefficient of Zn^2+^ can be estimated from the slope of the line using Equations (1) and (2) with the symbols explained in [42]. In particular, *n* is the number of electrons, *I_p_* is the peak current in A, *D* is the diffusion coefficient in cm^2^ s^−1^, *A* is the area of the electrode in cm^2^, *C* is the concentration of the electrolyte being 1 M, *ν* is the scan rate in V s^−1^, and α is the slope obtained from Figure 2c. It was found to be 4.85 × 10^−15^ cm^2^ s^−1^ for both cathodic and anodic processes. This value was comparable to other anodes for Li^+^ processes, including TiNb_2_O_7_ (1.16 × 10^−14^ cm^2^ s^−1^) [43], TiNb_6_O_17_ (5.35 × 10^−14^ cm^2^ s^−1^) [43], and nano-CaO-SnO_2_ (1 × 10^−14^ cm^2^ s^−1^) [44]. In contrast, it was found to be lower than those reported for layered iron vanadate (10^−11^−10^−10^ cm^2^ s^−1^) [45], zinc vanadium oxide (10^−10^−10^−9^ cm^2^ s^−1^) [46], Li^+^ intercalated V_2_O_5_·nH_2_O (0.95 × 10^−8^ cm^2^ s^−1^ to 3.37 × 10^−8^ cm^2^ s^−1^ during the discharge processes and 0.33 × 10^−8^ to 2.36 × 10^−8^ cm^2^ s^−1^ during the charge processes) [47], and porous framework zinc pyrovanadate (10^−10^ to 10^−9^ cm^2^ s^−1^) [48] in Zn^2+^ aqueous solutions. This comparison clearly shows that open-framework structures and/or the addition of graphene accelerates the Zn^2+^ diffusion process, indicating that there is space for improvement in the performance of (PEA)_3_Bi_2_I_9_.
(1)$$I_{p}=D^{1/2}2.72\times 10^{5}n^{3/2}AC\nu^{1/2}$$
(2)$$D^{1/2}=\frac{a}{2.72\times 10^{5}n^{3/2}AC}$$

The galvanostatic charge/discharge profiles of (PEA)_3_Bi_2_I_9_ were obtained at 0.4 A g^−1^ for the first and the 50th scan for a potential range of −0.5 V to +1.5 V, as illustrated in Figure 3a. There were two clear voltage plateaus in the discharge process (positions 1 and 2) and one plateau in the charge process, which were in agreement with the CV analysis. It is interesting to note that the voltage plateaus were well maintained upon the 50-cycling operations. The (PEA)_3_Bi_2_I_9_ could deliver a reversible specific capacity of approximately 220 mAh g^−1^ at 0.4 A g^−1^ with excellent stability after 50 scans and a capacity retention of almost 100%. Figure 3b,c presents the specific capacity at specific currents of 0.4 A g^−1^, 0.6 A g^−1^, and 0.7 A g^−1^ with excellent stability (i.e., the shape of the curves remained unchanged) and the capacity retention as a function, with the scan numbers for each specific current reaching almost 100% for all values. In addition, Figure 3d confirms the structural stability of the film after 50 continuous Zn^2+^ intercalation/deintercalation scans. We need to note that a similar pattern was observed after Zn^2+^ intercalation. In addition, the film was smoother with similar features (Figure 3d inset) as the pristine sample in Figure 1a.

Based on the electrochemical data shown above and the fact that XRD analysis indicated no shift of the expected peaks after Zn^2+^ intercalation, one may conclude that there was no significant change in the (PEA)_3_Bi_2_I_9_ structure. In this case, the observed redox peaks can be assigned to the reversible Zn^2+^ intercalation/deintercalation processes as shown below.
(PEA)_3_Bi_2_I_9_ + xZn^2+^ + xe^-^ ↔ (PEA)_3_Bi_2_I_9_Zn_x_

Until this report, there were no related reports of perovskites as anodes for aqueous Zn^2+^ batteries. However, perovskites such as CH_3_NH_3_PbBr_3_ [26,49], CsPb_2_Br_5_ [50], and Cs_4_PbBr_6_ [34] have been utilized in Li^+^ batteries as anode materials with good performance. In particular, CH_3_NH_3_PbBr_3_ presented a discharge capacity of 121 mAh g^−1^ after 200 continuous Li^+^ continuous intercalation/deintercalation scans [26]. CsPb_2_Br_5_ was reported to show a specific discharge capacity of 377 mAh g^−1^ with a retention of 75% after 100 scans [34]. In terms of Cs_4_PbBr_6_, the particular anode had a superior cyclic stability with the same specific capacity value of 549 mAh g^−1^ from the 100th to 1500th scan due to the large interfacial area between the perovskite material and the electrolyte [50]. Hence, the perovskite-based anodes described in this work suggested similar behavior to those reported in Li^+^ systems, but with a higher capacity value compared to CH_3_NH_3_PbBr_3_.

Overall, we may suggest that the stability and capacity of the film can be further improved through the addition of graphene and/or the alteration of processing parameters via the inclusion of an extra precursor or the replacement of the current substrate with a porous one to modify the morphology of the film or with another type such as Cu and stainless steel. These are issues that are under investigation in our lab.

## 4. Conclusions

It was possible to investigate thin films of (PEA)_3_Bi_2_I_9_ grown via AACVD as the anode in aqueous Zn^2+^ electrolytes as a proof of concept, with a specific capacity of 220 mAh g^−1^ at 0.4 A g^−1^ and presenting a capacity retention of almost 100% after 50 scans. Following the evaluation of cyclic voltammogram curves and structural analysis, the observed redox peaks can possibly be assigned to the reversible intercalation/deintercalation processes. The high capacity, low fabrication and operation costs, and the improved safety make the anode promising to utilize further in Zn^2+^ batteries.

## Figures and Tables

**Figure 1 nanomaterials-11-00656-f001:**
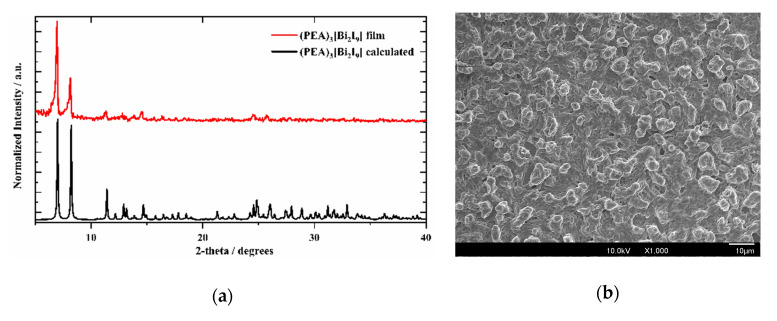
(**a**) Grazing Incidence X-ray diffraction (GIXRD) pattern of deposited (PEA)_3_Bi_2_I_9_ film and standard powder XRD pattern calculated from single crystal XRD. (**b**) Top surface SEM image of (PEA)_3_Bi_2_I_9_ film. Augmentation for the front view image is ×2000.

**Figure 2 nanomaterials-11-00656-f002:**
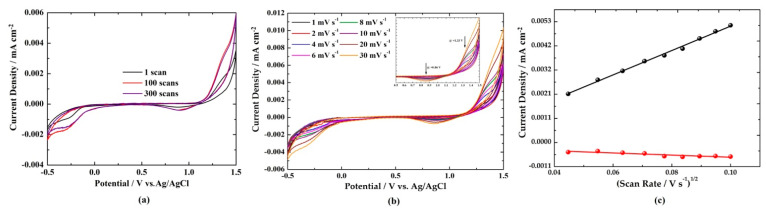
(**a**) Cyclic voltammogram curves of (PEA)_3_Bi_2_I_9_ for a scan rate of 10 mV s^−1^ and potential ranging −0.5 V to +1.5 V. (**b**) Cyclic voltammogram curves for different scan rates ranging from 1 to 30 mV s^−1^ and a magnified region of +0.5 V to +1.5 V as inset. (**c**) Linear fit of current density versus square root of scan rate at +1.35 V (cathode—black color) and +0.88 V (anode—red color). In all cases, an aqueous solution of 1 M ZnSO_4_^·^7H_2_O acts as an electrolyte.

**Figure 3 nanomaterials-11-00656-f003:**
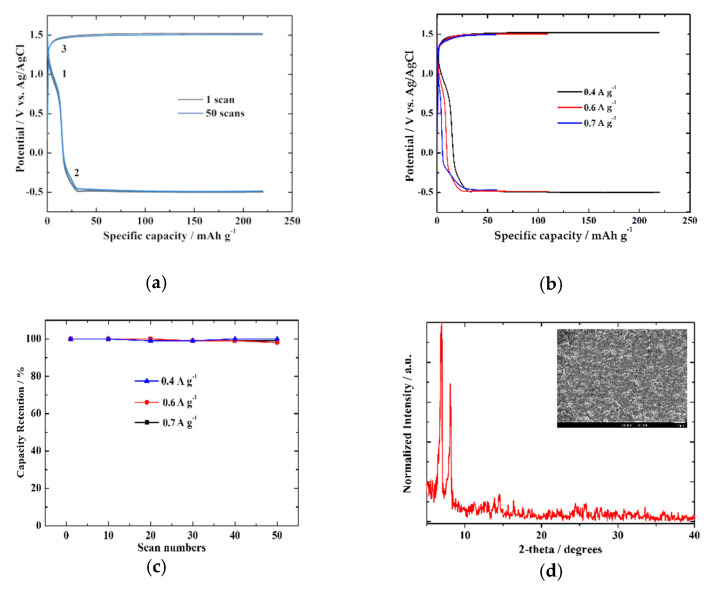
(**a**) Chronopotentiometric curves of (PEA)_3_Bi_2_I_9_ at a specific current of 0.4 A g^−1^ and a potential ranging of −0.5 V to +1.5 V. (**b**) Chronopotentiometric curves of the same anode material at 0.4 A g^−1^, 0.6 A g^−1^, and 0.7 A g^−1^. (**c**) Variation of the specific capacity with scan numbers at 0.4 A g^−1^, 0.6 A g^−1^, and 0.7 A g^−1^. In all cases, an aqueous solution of 1 M ZnSO_4_·7H_2_O acts as an electrolyte. (**d**) XRD pattern and SEM image (inset) of (PEA)_3_Bi_2_I_9_ film after 50 continuous Zn^2+^ intercalation/deintercalation scans.

## Data Availability

Data is contained within the article.

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
