# Peer review of "Electrochemical Investigation of Phenethylammonium Bismuth Iodide as Anode in Aqueous Zn2+ Electrolytes"

_nanomaterials, 2021, doi:10.3390/nano11030656_

Round 1

Reviewer 1 Report

This paper deals with the synthesis and electrochemical characterization in Zn2+ electrolyte of anode based on bismuth derivative. It is an interesting work, well written, concise, with a very rich bibliography.  However, the introduction of an analysis post mortem of these electrodes could improve the impact of this work.

Author Response

Dear Reviewer 1,

Thank you very much for the response and helpful suggestions regarding our manuscript entitled:

Investigation of Phenethylammonium Bismuth Iodide as Anode in Aqueous Zn2+ Electrolytes

We have done major revisions in our revised manuscript and we are resubmitting our work hoping that we have fully complied with your recommendations. The highlighted revision encloses all changes in bold and underline.

Being confident that we have addressed all of the remarks raised, we kindly ask you to reconsider our work for publication in Nanomaterials.

Reviewer’s comments:

This paper deals with the synthesis and electrochemical characterization in Zn2+ electrolyte of anode based on bismuth derivative. It is an interesting work, well written, concise, with a very rich bibliography. However, the introduction of an analysis post mortem of these electrodes could improve the impact of this work.

Thank you very much for the outcome. In the revised manuscript, you can find XRD and SEM analysis of the perovskite sample after 50 continuous Zn2+ intercalation/deintercalation scans in Figure 3 (d) with the necessary discussion in the Results and Discussion and the legend of Figure 3.

With our best regards,

Dimitra Vernardou, PhD and co-authors

Reviewer 2 Report

There are still some objections from my previous report that were not addressed by the authors.

Al the symbols in the equations 1 and 2 must be explained in the manuscript, not just cited. What was a concentration of Zn (C)used for calculation of diffusion coefficient and how was determined? 

In contrary to explaining sentence added  to the abstract and conclusions, I do not thing that considerable weighing error was taken into consideration, as far as reported charge capacity of 220 mAh/g is concerned. This value of charge capacity is rather rough estimate due to extremely low mass of active material (12 thousands of mg!!!!), the experimental error in specific charge capacities determination can be of the order of tens of percent! Since reported charge capacities are doubtful, the main goal of the study is doubtful as well. It can be presented as a qualitative proof of concept, but for the sake of scientific ethics and correctness, all this must be clearly stated in the manuscript.

Author Response

Dear Reviewer 2,

Thank you very much for the response and helpful suggestions regarding our manuscript entitled:

 Investigation of Phenethylammonium Bismuth Iodide as Anode in Aqueous Zn2+ Electrolytes

We have done major revisions in our revised manuscript and we are resubmitting our work hoping that we have fully complied with your recommendations. The highlighted revision encloses all changes in bold and underline.

Being confident that we have addressed all of the remarks raised, we kindly ask you to reconsider our work for publication in Nanomaterials.

Reviewer’s comments:

There are still some objections from my previous report that were not addressed by the authors.

Thank you very much for the outcome. Please see our response to your comments below.

1) Al the symbols in the equations 1 and 2 must be explained in the manuscript, not just cited. What was a concentration of Zn (C) used for calculation of diffusion coefficient and how was determined? 

Thank you for the comment. All symbols in equations 1 and 2 are explained along with the value of the electrolyte concentration in Results and Discussion section.

Regarding the electrolyte concentration, it was determined based on the amount required of zinc sulfate to dissolve in 100 ml deionized water for the preparation of 1 M aqueous solution (i.e. ZnSO4 + H2O -> Zn2+ + SO4-2). The reaction of zinc sulfate and water is included in Materials Characterization section.

2) In contrary to explaining sentence added  to the abstract and conclusions, I do not thing that considerable weighing error was taken into consideration, as far as reported charge capacity of 220 mAh/g is concerned. This value of charge capacity is rather rough estimate due to extremely low mass of active material (12 thousands of mg!!!!), the experimental error in specific charge capacities determination can be of the order of tens of percent! Since reported charge capacities are doubtful, the main goal of the study is doubtful as well. It can be presented as a qualitative proof of concept, but for the sake of scientific ethics and correctness, all this must be clearly stated in the manuscript.

Thank you for the comment. We clarified in both Abstract and Conclusions that this is a proof of concept.

With our best regards,

Dimitra Vernardou, PhD and co-authors

Reviewer 3 Report

The article entitled Electrochemical Investigation of Phenethylammonium Bismuth Iodide as Anode in Aqueous Zn2+ Electrolytes is worthy to be accepted in Nanomaterials after minor revision.

Here my comments:

1) What happens if phenyl ammonium in the structure is replaced by another ammonium ion?

2) The authors should add SEM image and EDS mapping of the (PEA)3[Bi2I9] film before and after cycling. Moreover a comparison between XRD patterns of (PEA)3[Bi2I9] film before cycling vs after cycling is requested

3) The authors must add a schematic that resumes the mechanism of reaction between (PEA)3[Bi2I9] and zinc ions

4) I suggest to make the figures more visible: it`s difficult to read the labels

Author Response

Dear Reviewer 3,

Thank you very much for the response and helpful suggestions regarding our manuscript entitled:

 Investigation of Phenethylammonium Bismuth Iodide as Anode in Aqueous Zn2+ Electrolytes

We have done major revisions in our revised manuscript and we are resubmitting our work hoping that we have fully complied with your recommendations. The highlighted revision encloses all changes in bold and underline.

Being confident that we have addressed all of the remarks raised, we kindly ask you to reconsider our work for publication in Nanomaterials.

Reviewer’s comments:

The article entitled Electrochemical Investigation of Phenethylammonium Bismuth Iodide as Anode in Aqueous Zn2+ Electrolytes is worthy to be accepted in Nanomaterials after minor revision.

Thank you for the outcome. Please see below our response to your comments.

1) What happens if phenyl ammonium in the structure is replaced by another ammonium ion?

Thank you for the comment. There are reports of using methylammonium bismuth iodide (see e.g. https://www.sciencedirect.com/science/article/pii/S2211285518303197?via%3Dihub). Nevertheless, the phenethylammonium cation was chosen in this work since it provides enhanced stability and resistance against moisture. The respective information was included in Introduction.

2) The authors should add SEM image and EDS mapping of the (PEA)3[Bi2I9] film before and after cycling. Moreover a comparison between XRD patterns of (PEA)3[Bi2I9] film before cycling vs after cycling is requested

Thank you very much for the comment. We have included Figure 3 (d) indicating the XRD and SEM after 50 continuous Zn2+ intercalation/deintercalation scans.

3) The authors must add a schematic that resumes the mechanism of reaction between (PEA)3[Bi2I9] and zinc ions

Thank you for the comment. In the revised manuscript, there is a proposed reaction regarding the reversible Zn2+ intercalation/deintercalation processes in the anode based on the data obtained.

4) I suggest to make the figures more visible: it`s difficult to read the labels

Thank you for the comment. We modified the contrast in the Figures.

With our best regards,

Dimitra Vernardou, PhD and co-authors

Round 2

Reviewer 2 Report

I have no more comments or suggestions.

This manuscript is a resubmission of an earlier submission. The following is a list of the peer review reports and author responses from that submission.

Round 1

Reviewer 1 Report

This communication reported the phenethylammonium bismuth iodide as anode for Zinc-ion battery. Physical and electrochemical characterizations are provided. However, some issues should be addressed before its publication.

In the introduction section, the authors should give a description of the Zn anode and its improvement strategy. 

Has the author ever matched any cathode material with this anode? What is the capacity and cycling performance?

I suggest that the author explore the influence of electrolyte type on the performance of anode modification in future research.

Author Response

Dear Reviewer 1,

Thank you very much for your response and helpful suggestions regarding the manuscript entitled:

Investigation of Phenethylammonium Bismuth Iodide as Anode in Aqueous Zn2+ Electrolytes

we wish to publish in Nanomaterials. We have done major revisions in our original manuscript and we are resubmitting our work hoping that we have fully complied with your recommendations. The highlighted revision encloses all changes in bold and underline.

Reviewer’s comments:

This communication reported the phenethylammonium bismuth iodide as anode for Zinc-ion battery. Physical and electrochemical characterizations are provided. However, some issues should be addressed before its publication.

Thank you very much for your outcome. Please see below our response to your comments.

  1. In the introduction section, the authors should give a description of the Zn anode and its improvement strategy. 

Thank you for the comment. In Introduction, we have included additional information on Zn anode and its improvement strategy along with the necessary references.

  1. Has the author ever matched any cathode material with this anode? What is the capacity and cycling performance?

Thank you for the comment. At this stage, the main objective and priority of this work was the investigation of the electrochemical performance of phenethylammonium bismuth iodide. This is the first time that such an investigation is reported for this type of material, which presents unique properties with a specific capacity of 220 mAh g-1 at 0.4 A g-1 with excellent stability after 50 scans.

Undoubtedly, the characterization of electrode materials using a Zn metal counter electrode is quite important and we consider this as the next step for further exploitation of our electrodes in Zn2+ batteries technology.

In order to make clear this point, we altered the manuscript title to “Electrochemical Investigation of Phenethylammonium Bismuth Iodide as Anode in Aqueous Zn2+ Electrolytes”

  1. I suggest that the author explore the influence of electrolyte type on the performance of anode modification in future research.

Thank you for the suggestion. At the beginning of the Electrochemical Studies section, we have added additional information to show all the work carried out with the electrolyte concentration prior the cyclic voltammetry analysis.

With our best regards,

Dimitra Vernardou, PhD and co-authors

Reviewer 2 Report

The manuscript under review deals with study of phenethylammonium bismuth organohalide as an anode for aqueous Zn2+ batteries. The authors claim reasonable charge capacity and excellent stability and capacity retention of the anode, but there are several contradictory issues and discrepancies, specified below, to be addressed.  My recommendation is major revision of the manuscript.

  1. The authors discus Zn2+ intercalation/deintercalation to the (PEA)3[Bi2I9] structure. Where exactly in its structure are Zn2+ cations accommodated and what is a concentration of Zn in fully charged anode? Which particular redox processes take place at potentials corresponding with current peak maximum?
  2. Kinetic analysis and discussion about diffusion controlled and capacitive mechanism of charge storage are confusing. According to Figure 2c plotting current density in the peak maximum vs. square root of the scan rate, both dependencies at 1.35V and 0.88V are perfectly linear, so how can be the exponent calculated according to the equation (3) 0.72 for the peak at 0.88V? How can be value of the exponent from the equation (3) lower than 0.5, as it is presented for the peak at -0.27V? From the equation (1) in the manuscript and the equation |iC|= ACdv, describing dependence of capacitive current vs. scan rate it is obvious that capacitive charge storage process is faster, hence, its contribution to the overall current is dominating at higher scan rates.

        Could the authors explain which particular process of charge storage is            addressed in the sentence below?

        However, the b value of the peak at -0.27 V is lower than 0.5                          suggesting a diffusion-limited process as confirmed especially at                      high scan rates.

        Generally, all the voltammetric peaks in Figures 2a, b are not well                    developed, so I doubt if it is possible unequivocally determine the                    current peak maximum.

  1. The authors ascribe increase of the specific charge capacity with current density at galvanostatic chronopotentiometry to fast diffusion kinetics of Zn2+, which could be supported by CVs at higher scan rates. This statement is difficult to understand. At cyclic voltammetry, current at higher scan rates is higher, but the time of charging is shorter. Since the charge is calculated as current (I) multiplied by time (t), maximum charge capacity of the material is reached at slower scan rates.

At galvanostatic measurements, higher charging current increases overpotential and leads to lower charge capacity of the battery.

There are additional discrepancies in the part discussing galvanostatic chronopotentiometry and in Figure 3. In Figure 2, all the currents are normalized to the area of thin film electrode. In Figure 3, all the capacities are normalized to the mass of active material. How did the authors determine the mass of active material on a thin film electrode? In addition, specific capacity for all the current densities presented in Figure 3c is the same-100mAh/g.

Some of sentences are practically word-for-word copied from earlier paper of the corresponding author, just electrode material is changed, se below:

This work:

The  upward increase of the capacity with increasing specific current (i.e. 110 mAh g-1 at 0.2 A g-1 and 59 mAh g-1 at 0.1 A g-1) can be due to the fast diffusion kinetics of Zn2+ into the (PEA)3[Bi2I9], which is also supported by the CV curves at different scan rates.

Ref. 49:

The upward trend of the specific discharge capacity with increasing specific current can be due to the fast diffusion kinetics of Mg2+into the metal oxide framework. This is also supported by cyclic voltammetry measured at increasing scan rates (Fig. 3B).

  1. In conclusion, I doubt, whether thin film electrode on a conducting glass is applicable as an anode in batteries. The mass and thickness of conducting glass as a current collector substantially decreases both the battery energy density and volumetric density, respectively.

Author Response

Dear Reviewer 2,

Thank you very much for your response and helpful suggestions regarding the manuscript entitled:

Investigation of Phenethylammonium Bismuth Iodide as Anode in Aqueous Zn2+ Electrolytes

we wish to publish in Nanomaterials. We have done major revisions in our original manuscript and we are resubmitting our work hoping that we have fully complied with your recommendations. The highlighted revision encloses all changes in bold and underline.

Reviewer’s comments:

The manuscript under review deals with study of phenethylammonium bismuth organohalide as an anode for aqueous Zn2+ batteries. The authors claim reasonable charge capacity and excellent stability and capacity retention of the anode, but there are several contradictory issues and discrepancies, specified below, to be addressed.  My recommendation is major revision of the manuscript.

We would like to thank you for the constructive remarks, which help us to improve our manuscript. Please see below our response to your concerns.

  1. The authors discuss Zn2+ intercalation/deintercalation to the (PEA)3[Bi2I9] structure. Where exactly in its structure are Zn2+ cations accommodated and what is a concentration of Zn in fully charged anode? Which particular redox processes take place at potentials corresponding with current peak maximum?

Thank you for the comment. We would like to clarify that the main objective and priority of this work was the investigation of the electrochemical performance of phenethylammonium bismuth iodide. This is the first time that such an investigation is reported for this type of material, which presents unique properties.

Undoubtedly, the in-situ or ex-situ characterization of electrode materials during the intercalation/deintercalation scans is quite important and we consider this as the next step for further exploitation of our electrodes in Zn2+ batteries technology.

Based on the electrochemical data and the fact that XRD analysis indicated no shift of the expected peaks after Zn2+ intercalation, one may say that there is no significant change of the (PEA)3[Bi2I9] structure. In this case, the observed redox peaks can be assigned to a reversible Zn2+ intercalation/deintercalation processes as shown below

(PEA)3[Bi2I9] + xZn2+ + 2xe- ↔ (PEA)3[Bi2I9]Znx

The respective changes have been performed in Results and Discussion to indicate the point discussed above.

We have also changed the manuscript title to “Electrochemical Investigation of Phenethylammonium Bismuth Iodide as Anode in Aqueous Zn2+ Electrolytes” to make clear the objective of the work.

  1. Kinetic analysis and discussion about diffusion controlled and capacitive mechanism of charge storage are confusing. According to Figure 2c plotting current density in the peak maximum vs. square root of the scan rate, both dependencies at 1.35V and 0.88V are perfectly linear, so how can be the exponent calculated according to the equation (3) 0.72 for the peak at 0.88V? How can be value of the exponent from the equation (3) lower than 0.5, as it is presented for the peak at -0.27V? From the equation (1) in the manuscript and the equation |iC|= ACdv, describing dependence of capacitive current vs. scan rate it is obvious that capacitive charge storage process is faster, hence, its contribution to the overall current is dominating at higher scan rates.

Could the authors explain which particular process of charge storage is addressed in the sentence below?

However, the b value of the peak at -0.27 V is lower than 0.5 suggesting a diffusion-limited process as confirmed especially at high scan rates.

Generally, all the voltammetric peaks in Figures 2a, b are not well developed, so I doubt if it is possible unequivocally determine the current peak maximum.

Thank you for the comment. We realize that the kinetic studies of our data cannot be included. The discussion and the graphs related with the power law have been removed. Further measurements and analysis will be performed in our future work.

The order of magnitude of current density in cyclic voltammograms is low (i.e. 0.006 mA/cm2), which however does not reduce the importance of our data. In order to support our data, we have modified Figure 2(b) and the text in the manuscript accordingly.

  1. The authors ascribe increase of the specific charge capacity with current density at galvanostatic chronopotentiometry to fast diffusion kinetics of Zn2+, which could be supported by CVs at higher scan rates. This statement is difficult to understand. At cyclic voltammetry, current at higher scan rates is higher, but the time of charging is shorter. Since the charge is calculated as current (I) multiplied by time (t), maximum charge capacity of the material is reached at slower scan rates.

At galvanostatic measurements, higher charging current increases overpotential and leads to lower charge capacity of the battery.

There are additional discrepancies in the part discussing galvanostatic chronopotentiometry and in Figure 3. In Figure 2, all the currents are normalized to the area of thin film electrode. In Figure 3, all the capacities are normalized to the mass of active material. How did the authors determine the mass of active material on a thin film electrode? In addition, specific capacity for all the current densities presented in Figure 3c is the same-100mAh/g.

Some of sentences are practically word-for-word copied from earlier paper of the corresponding author, just electrode material is changed, se below:

This work:

The  upward increase of the capacity with increasing specific current (i.e. 110 mAh g-1 at 0.2 A g-1 and 59 mAh g-1 at 0.1 A g-1) can be due to the fast diffusion kinetics of Zn2+ into the (PEA)3[Bi2I9], which is also supported by the CV curves at different scan rates.

Ref. 49:

The upward trend of the specific discharge capacity with increasing specific current can be due to the fast diffusion kinetics of Mg2+ into the metal oxide framework. This is also supported by cyclic voltammetry measured at increasing scan rates (Fig. 3B).

Thank you for the comment. We have repeated the galvanostatic measurements for specific current 0.6 and 0.7 A g-1 indicating α decrease of specific capacity for higher specific current. We believe it was a numerical error. The corresponding changes have been performed in the text, Figure 3 (b) and the Figure caption.

In cyclic voltammetry, the current density (and not specific current) was utilized for the estimation of the diffusion coefficient. In order to avoid the confusion, all y-axes were converted into Current / mA instead Current / mA cm-2. There are no changes in the current values since the area of the anode electrode was 1 cm2.

The mass of the anode electrode was estimated by a 6-digit analytical grade scale measuring the FTO glass substrate before and after the growth. It was found to be 0.000012 g. The specific information is included in Materials Characterization section.

In Figure 3 (c), y-axis is Capacity Retention / % and not Specific Capacity / mAh g-1.

Finally, the sentences that are similar with previous work have been removed.

  1. In conclusion, I doubt, whether thin film electrode on a conducting glass is applicable as an anode in batteries. The mass and thickness of conducting glass as a current collector substantially decreases both the battery energy density and volumetric density, respectively.

Thank you for the comment. In our lab, we are currently working on the growth of the anode electrode on copper and stainless steel. We have added this information in the last paragraph of Results and Discussion section.

With our best regards,

Dimitra Vernardou, PhD and co-authors

Reviewer 3 Report

This manuscript is described about phenethylammonium bismuth iodide anode for Zn battery.

Their experiment procedure is not good to prove performance of the anode material.

They cut the voltage 1.5 V however, from CV curve, the oxidation reaction is still ongoing. They cannot evaluate the capacity of the anode correctly in this experiment.

Author Response

Dear Reviewer 3,

Thank you very much for your response and helpful suggestions regarding the manuscript entitled:

Investigation of Phenethylammonium Bismuth Iodide as Anode in Aqueous Zn2+ Electrolytes

we wish to publish in Nanomaterials. We have done major revisions in our original manuscript and we are resubmitting our work hoping that we have fully complied with your recommendations. The highlighted revision encloses all changes in bold and underline.

Reviewer’s comments:

This manuscript is described about phenethylammonium bismuth iodide anode for Zn battery. Their experiment procedure is not good to prove performance of the anode material. They cut the voltage 1.5 V however, from CV curve, the oxidation reaction is still ongoing. They cannot evaluate the capacity of the anode correctly in this experiment.

First of all, we would like to thank you for the comments. At this stage, the main objective and priority of the work was the investigation of the electrochemical performance of phenethylammonium bismuth iodide. This is the first time that such an investigation is reported for this type of material, which presents unique properties with a specific capacity of 220 mAh g-1 at 0.4 A g-1 and an excellent stability after 50 scans. The investigation of the electrochemical performance is the first important test towards the identification of suitable materials for Zn2+ battery electrodes.

In order to make clear this point, we altered the manuscript title to “Electrochemical Investigation of Phenethylammonium Bismuth Iodide as Anode in Aqueous Zn2+ Electrolytes”.

Starting from lab-scale results, we have performed meticulous electrochemical studies including the stability window and the electrolyte concentration to find the best performing electrode-electrolyte combination. A similar electrochemical testing procedure has been adopted in several works found in the literature (See for example, Kim et al., Adv. Mater. 2015, 27, 2015, 3377-3384; Wang et al. ACS Appl. Mater. Interfaces, 2018, 10 (8), 7061–7068; Tan et al., J. Am. Chem. Soc. 2017, 139, 8772−8776).

At the beginning of the Electrochemical Studies section, details regarding the cyclic voltammetry experiments are included in order to explain how the electrochemical window was determined.

Undoubtedly, the characterization of electrode materials using a Zn metal counter electrode is quite important. However we consider this as the next step for further exploitation of our electrodes in Zn2+ batteries technology.

To further clarify the main objective of this work, as explained above, the last paragraph of the Introduction has been modified accordingly.

With our best regards,

Dimitra Vernardou, PhD and co-authors

Reviewer 4 Report

This paper shows interesting results on zink battery research which definitely is a good approach, going toward more convenient materials.

I understand this line of research is at the beginning, still though enough is known for you to sum up the background a little better in the introduction. The information given is not very clear.

What are typical conductivity values for these systems? What is happening on the interface? Do you have some idea about thermal stability?

line 178 You say that the perovskite-based anodes shown in this work suggest comparable behaviour with those reported in Li+ systems with an enhanced performance in some cases. / this is a very weak statement....in which cases???

In conclusion, only your results should be summed up future plans should be mover to the end of results and discussion.

The level of English is not good enough, please consider revision.

Author Response

Dear Reviewer 4,

Thank you very much for your response and helpful suggestions regarding the manuscript entitled:

Investigation of Phenethylammonium Bismuth Iodide as Anode in Aqueous Zn2+ Electrolytes

we wish to publish in Nanomaterials. We have done major revisions in our original manuscript and we are resubmitting our work hoping that we have fully complied with your recommendations. The highlighted revision encloses all changes in bold and underline.

Reviewer’s comments:

This paper shows interesting results on zinc battery research which definitely is a good approach, going toward more convenient materials.

Thank you very much for your outcome. Please see below our response to your comments.

  1. I understand this line of research is at the beginning, still though enough is known for you to sum up the background a little better in the introduction. The information given is not very clear.

Thank you for the comment. In Introduction, we have included information for the Zn anode characteristics and the strategies undertaken to tackle the associated challenges. In addition, we have added a few sentences at the end of this section to clarify the objective of the work.

  1. What are typical conductivity values for these systems? What is happening on the interface? Do you have some idea about thermal stability?

Thank you for the comment. We have not measured the conductivity of our system. In addition, it has not been reported in the literature to the best of our knowledge. We will make efforts to measure it on our next publication.

We would like to clarify that the main objective and priority of this work was the investigation of the electrochemical performance of phenethylammonium bismuth iodide. Undoubtedly, the in-situ or ex-situ characterization of electrode materials during the intercalation/deintercalation scans is quite important and we consider this as the next step for further exploitation of our electrodes in Zn2+ batteries technology.

This perovskite material could keep stable up to 300 oC. Please see also J. Mater. Chem. 2019, 7, 20733. The respective information has been added in Introduction.

  1. line 178 You say that the perovskite-based anodes shown in this work suggest comparable behaviour with those reported in Li+ systems with an enhanced performance in some cases. / this is a very weak statement....in which cases???

Thank you for the comment. This sentence has been rephrased to make it clearer.

  1. In conclusion, only your results should be summed up future plans should be mover to the end of results and discussion.

Thank you for the comment. We have modified Conclusions according to your suggestion.

  1. The level of English is not good enough, please consider revision.

Thank you for the comment. We have made a thorough English revision in our manuscript.

With our best regards,

Dimitra Vernardou, PhD and co-authors

Round 2

Reviewer 2 Report

Generally, I am satisfied with authors’ corrections of the manuscript. It has been improved a lot. However, there are still several issues to be addressed.

  1. In Figure 2, current density should be expressed in mA/cm2, not in mA, so the label of vertical axes in Figures 2a b, c should be the same as in the previous version of the manuscript.
  2. The symbols in the equations 1 and 2 are not explained in the manuscript. What was a concentration of Zn used for calculation of diffusion coefficient? Reference 42 is wrong, neither eq 1 or 2 are mentioned in ref. 42.
  3. Since the amount of active material on the electrode, used for calculation of specific capacity from galvanostatic measurements, is extremely low (0.000012 g), resulting charge capacities values are strongly affected by weighing error. This must be stated in the manuscript alongside the statement about charge capacities, i.e. in abstract and in conclusions.

Reviewer 3 Report

Their voltage range in the charge and discharge test included oxidative decomposition of the electrolyte. This caused large error of capacity of electrode material. Therefore, I strongly recommend the authors consider their experimental condition.